# MS-Former: Multi-Scale Self-Guided Transformer for Medical Image Segmentation

**Sanaz Karimijafarbigloo** [1,2]          sanaz.karimijafarbloo@ur.de

**Reza Azad** [2]          Reza.Azad@lfb.rwth-aachen.de

**Amirhossein Kazerouni** [3]          amirhossein477@gmail.com

**Dorit Merhof** [1,4]          dorit.merhof@ur.de

[1] *Faculty of Informatics and Data Science, University of Regensburg, Regensburg, Germany*

[2] *Institute of Imaging and Computer Vision, RWTH Aachen University, Aachen, Germany*

[3] *School of Electrical Engineering, Iran University of Science and Technology, Tehran, Iran*

[4] *Fraunhofer Institute for Digital Medicine MEVIS, Bremen, Germany*

**Editors:** Accepted for publication at MIDL 2023

## Abstract

Multi-scale representations have proven to be a powerful tool since they can take into account both the fine-grained details of objects in an image as well as the broader context. Inspired by this, we propose a novel dual-branch transformer network that operates on two different scales to encode global contextual dependencies while preserving local information. To learn in a self-supervised fashion, our approach considers the semantic dependency that exists between different scales to generate a supervisory signal for inter-scale consistency and also imposes a spatial stability loss within the scale for self-supervised content clustering. While intra-scale and inter-scale consistency losses aim to increase features similarly within the cluster, we propose to include a cross-entropy loss function on top of the clustering score map to effectively model each cluster distribution and increase the decision boundary between clusters. Iteratively our algorithm learns to assign each pixel to a semantically related cluster to produce the segmentation map. Extensive experiments on skin lesion and lung segmentation datasets show the superiority of our method compared to the state-of-the-art (SOTA) approaches. The implementation code is publicly available at GitHub.

**Keywords:** Transformer, Inter-scale, Intra-scale, Segmentation, Medical Image.

## 1. Introduction

Over the past few years, there has been a remarkable success of supervised deep learning methods in computer vision tasks given a large-scale annotated dataset for several downstream tasks such as image classification (Mangalam et al., 2022; Dong et al., 2022), semantic segmentation (You et al.; Heidari et al., 2023), object detection (Ren et al., 2015; He and Todorovic, 2022), etc. Nonetheless, the collection and annotation of a large-scale dataset are time-consuming, tedious, and expensive tasks, particularly for medical imaging datasets. While obtaining adequate good-quality annotated data is challenging, the unlabeled data is available in abundance. To alleviate the problem of annotated data scarcity, several approaches were proposed in the literature. Transfer learning, as a gold standard method in this direction, performs representational learning by fine-tuning the pre-trained network on the new task. Although the knowledge transfer provides a good starting point

for the optimization algorithm, the lack of annotated data on the downstream task limits the convergence of the network for learning task-specific features and results in less stable models. Moreover, concerning the predefined model architecture, this method seems to be inefficient in a complex task such as segmentation (Zhou et al., 2017; He et al., 2019; Araújo et al., 2022). Alternatively, unsupervised approaches reformulate the problem by learning data-driving features from the image itself (Ahn et al., 2019, 2020; Gao et al., 2022). However, the obtained results of these methods are not always reliable since no label or measure is available to confirm their efficiency (Khan et al., 2019). Semi-supervised approaches are another alternative that try to bridge between supervised and unsupervised learning to address data scarcity. They use a small portion of labeled data in conjunction with a large number of unlabeled data to train a predictive model (Luo et al., 2021, 2022). Even though the semi-supervised setting guides the learning process by a small amount of annotated data, in the case that the labeled data fails to represent the entire distribution this approach will be inefficient. Unlike the aforementioned strategies which perform the learning paradigm by modeling the data distribution, the self-supervised technique uses a different perspective by defining a set of matching tasks. More precisely, this strategy creates a supervisory signal from the image itself to perform representational learning. Zhuang et al. (Zhuang et al., 2019) proposed a 3D self-supervised learning approach for brain tumor segmentation. They trained a 3D CNN utilizing a novel proxy task and then fine-tuned the pre-trained weights on their specific tasks with manual labels. Zheng et al. (Zheng et al., 2021) proposed a hierarchical self-supervised learning to learn semantic features from multi-domain data for various medical image segmentation tasks. Bai et al. (Bai et al., 2019) proposed a network in a self-supervised manner to learn features by predicting anatomical positions in order to segment cardiac MR images. Tajbakhsh et al. (Tajbakhsh et al., 2019) took a different perspective and proposed a pretext task to predict color, rotation, and noise. They claimed that such surrogate supervision enables the network to learn data-driven and generic features which are cardinal for lung lobe segmentation and nodule detection tasks. Chen et al. (Chen et al., 2019) proposed a self-supervised learning network based on context restoration to segment brain tumors in multi-modal MRIs. Another work (Tao et al., 2020) has utilized a self-supervised learning framework for 3D medical image segmentation, where a volume-wise transformation is used to better exploit 3D anatomical information. Ahn et al. (Ahn et al., 2021) further extended the self-supervised technique without relying on the annotation data for the segmentation task. In their approach, a spatial-guided clustering method is presented to iteratively combine neighboring clusters and predict the segmentation mask. However, to learn long-range dependencies, there is no mechanism included in this approach. This might explain the limitation of this approach for capturing long-range dependency and merging clusters with shared characteristics, specifically in complex backgrounds. To address the above-mentioned limitations, we propose MS-Former, which models both local semantic dependencies and global context correlation for a semantic segmentation task. Our strategy utilized a dual-stream transformer block equipped with several self-supervisory signals to ensure feature consistency within the cluster while increasing the intra-clustering margin. In the next section, we will elaborate more on our strategy. Our contributions are: **1)** a novel pure Transformer model to impose hierarchical consistency loss in a self-supervised manner. **2)** Spatial and feature consistency

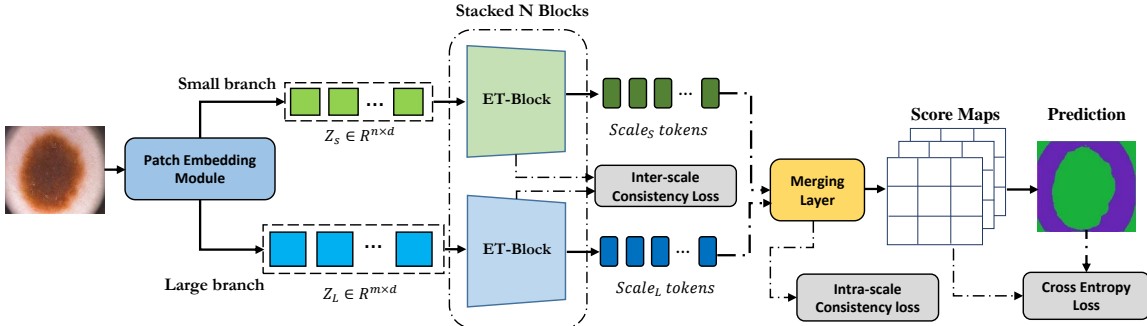

Figure 1: The overview of the proposed MMCFormer. In MMCFormer two vision transformer models are employed in parallel to capture multi-scale representation. The approach also utilizes inter-scale and intra-scale consistency losses to provide supervisory signals for feature matching and semantic learning, thereby enhancing the performance of the semantic segmentation task.

losses to learn a feature clustering space based on the same characteristics. **3)** SOTA results on skin lesion and lung segmentation challenges.

## 2. Proposed Method

A critical challenge in self-supervised semantic segmentation is how to guide the network to consider semantic and local dependencies for clustering each pixel into a set of shared characteristics. More specifically, we are interested in modeling the representational space in such a way that it is possible to learn the underlying distribution of the semantic and local dependencies among image regions and provide discriminative information for clustering each pixel. To address this issue, we propose a multi-scale self-guided Transformer network, MS-Former, to perform self-supervised semantic segmentation tasks without requiring any annotation label. The structure of the MS-Former is shown in Figure 1. Our network builds upon an efficient transformer block; hence, we first present the efficient transformer block, and then we will elaborate on our multi-scale self-supervised strategy.

### 2.1. Efficient Transformer Module

The main drawback of the vision transformer model is the quadratic computational complexity of the self-attention mechanism. More specifically, the standard self-attention block calculates the attention matrix using the query $(Q^{1 \times d})$ and key $(K^{1 \times d})$ values and then multiples with the value $(V^{1 \times d})$ vector to perform the normalization as follows:

$$S(\mathbf{Q}, \mathbf{K}, \mathbf{V}) = Softmax\left(\frac{\mathbf{Q}\mathbf{K^T}}{\sqrt{d}}\right)\mathbf{V} \tag{1}$$

where $n$ shows the number of tokens and $d$ indicates the embedding dimension. This operation has $O(n^2)$ complexity. Shen et al. (Shen et al., 2021) argue that the context

representation in the self-attention module contains redundant information and suggest a modified equation to reduce the computation burden into a linear form:

$$\mathbf{E}(\mathbf{Q}, \mathbf{K}, \mathbf{V}) = \rho_{\mathbf{q}}(\mathbf{Q}) \left( \rho_{\mathbf{k}}(\mathbf{K})^{\mathbf{T}} \mathbf{V} \right) \tag{2}$$

In their strategy, first, the Softmax function denoted as $\rho$ is applied to the key and query vectors to provide normalized scores. Next, the global context is formed by calculating the matrix multiplication between the key and value. Statistically, they show that efficient attention produces an equivalent representation while staying computationally linear in terms of the number of tokens $O(d^2 n)$. Figure 2(a) shows the efficient self-attention block in more details.

## 2.2. Network Architecture

Giving an input image $X^{H \times W \times C}$, where $H \times W$ shows the spatial dimension and $C$ refers to the number of channels, our network first applies the patch embedding module in two different scales, namely small $(P_s)$ and large $(P_l)$, to generate tokenized sequences $z_s^{n \times d}$, $z_l^{m \times d}$, where $n$ and $m$ show the number of non-overlapping windows in dimension $d$. We feed the generated sequences into a dual-branch transformer network which uses the same transformer structure in each path but operates on different input sequences to capture both coarse-grained and fine-grained features while staying computationally linear. We then concatenate the translated sequences in channel dimension and perform an $MLP$ down-sampling layer as follows:

$$\mathbf{z}' = \left[ f^l \left( \mathbf{z}_l \right) \| UP(f^s \left( \mathbf{z}_s \right)) \right], \ \mathbf{z} = \mathbf{MLP}(\mathbf{z}'), \tag{3}$$

where, $f^l$ and $f^s$ show the large and small branch networks, respectively. Also, $UP$ illustrates up-sampling, and $MLP$ is used to demonstrate the linear operations. We reshape the generated sequence $z$ into a soft prediction map $S^{H \times W \times K}$, where $K$ indicates the number of clusters. We then create the semantic segmentation map $Y^{H \times W \times K}$ by applying the argmax function on each spatial location to activate the related cluster index. We train the network by iteratively minimizing the cross-entropy loss between the soft prediction map and the segmentation map:

$$\mathcal{L}_{ce}(S, Y) = -\frac{1}{H \times W} \sum_{i=1}^{H \times W} \sum_{j=1}^{K} Y_{i,j} \log \left( S_{i,j} \right). \tag{4}$$

Although the cross-entropy loss included in our approach learns the underlying distribution of the clusters by continually increasing the network confidence for mapping similar pixels into the same cluster while increasing the inter-cluster margin, it lacks to model the spatial relationship in a local region and renders a poor performance for merging neighboring clusters. Besides that, the supervision only applies to the output channel, which for the segmentation task deep supervision is more critical to ensure feature consistency in each level of the network. To address these limitations, we propose to include intra-scale and inter-scale feature matching mechanisms to adaptively recalibrate the feature representation in the local neighborhood while imposing feature consistency in a multi-scale manner.

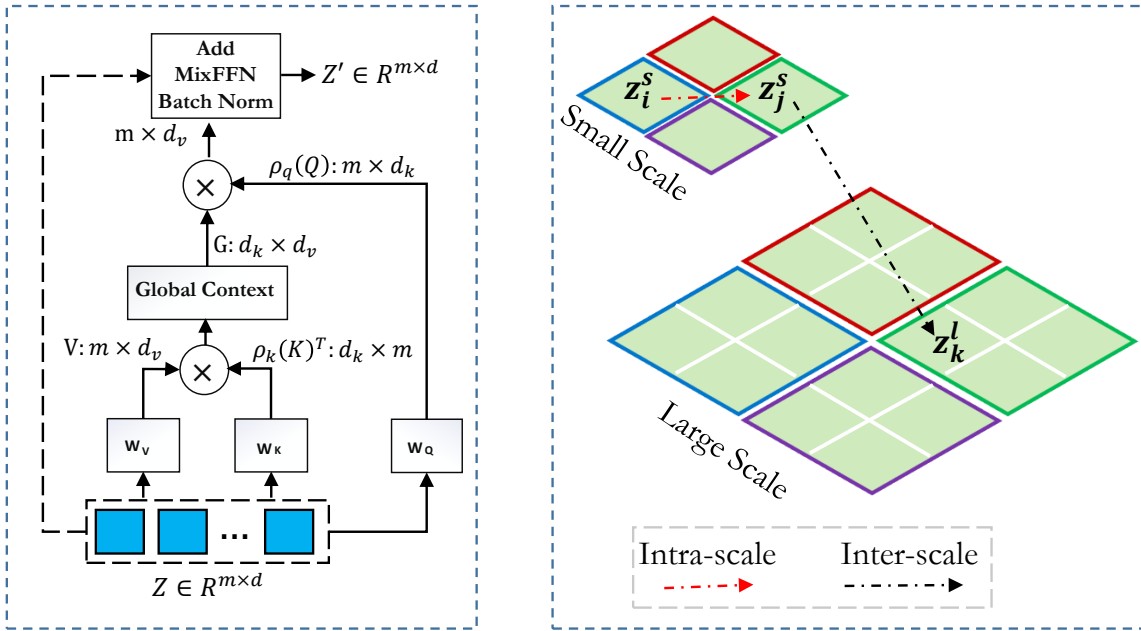

Figure 2: (a): structure of the efficient Transformer module and (b): visualization of the intra-scale and inter-scale dependencies.

## 2.3. Inter-scale Consistency

The use of multi-scale representations has been demonstrated to be an effective tool, as they allow for the incorporation of both the small details of the objects within an image and the larger contexts. When we create representations of the same object at different scales, the correlation of these representations should be similar because the nature of the object remains the same regardless of the scale at which it is represented. In light of this, we propose an inter-scale module that seeks to provide a supervisory signal to maximize the context correlation between small and large branches, thereby approaching these two branch distributions (Figure 2(b) illustrates the inter-scale concept). Our efficient self-attention mechanism calculates the global context using the query and value vectors as $\mathbf{G} = (\rho_k(k)^T V) \in R^{d \times d}$, where it shows the correlation matrix. We geometrically represent $\mathbf{G}^s$ and $\mathbf{G}^l$ as the global context of small and large branches, respectively. Hence, to align the distribution of both context vectors, we aim to maximize the correlation of these two distributions. We show in Equation 5 that maximizing the correlation between $\mathbf{G}^s$ and $\mathbf{G}^l$ can be viewed as minimizing the angle between them:

$$\cos(\mathbf{G^s}, \mathbf{G^l}) = \frac{\langle \mathbf{G^s}, \mathbf{G^l} \rangle}{\|\mathbf{G^s}\| \cdot \|\mathbf{G^l}\|} = \frac{\langle \mathbf{G^s}, \mathbf{G^l} \rangle}{\sqrt{\langle \mathbf{G^s}, \mathbf{G^s} \rangle} \cdot \sqrt{\langle \mathbf{G^l}, \mathbf{G^l} \rangle}}$$

$$= \frac{\mathbf{\Sigma_i^d \Sigma_j^d G_{ij}^s G_{ji}^l}}{\sqrt{\mathbf{\Sigma_i^d \Sigma_j^d G_{ij}^{s\,2}}} \sqrt{\mathbf{\Sigma_i^d \Sigma_j^d G_{ij}^{l\,2}}}} = \frac{\mathrm{Cov}\left(\mathbf{G^s}, \mathbf{G^l}\right)}{\sigma\left(\mathbf{G^s}\right)\sigma\left(\mathbf{G^l}\right)} \tag{5}$$

$$= \rho\left(\mathbf{G^s}, \mathbf{G^l}\right)$$

Therefore, we propose to minimize the cosine dissimilarity:

$$\mathcal{L}_{inter}(\mathbf{G}, y) = 1 - \cos\left(\mathbf{G}^s, \mathbf{G}^l\right). \tag{6}$$

$\mathcal{L}_{inter}$ allows for the exchange of inter-scale information, which can improve the representation of features and align the distribution of context vectors $\mathbf{G}^s$ and $\mathbf{G}^l$. Furthermore, this supervisory signal allows the network to better model the clustering space by grouping similar clusters in a multi-scale manner.

## 2.4. Intra-scale Consistency

The cross-entropy loss utilized in our network measures the difference between the clusters' probability distributions and provides a mechanism to assign similar pixels to the same distribution. However, applying the cross-entropy loss alone would not directly consider the spatial arrangement of the pixels or regions in the image. Therefore, we provide an intra-scale module to account for the spatial relationships and merge the clusters with similar characteristics. To this end, we create two tokenized sequences $(A_1^{n \times d}, A_2^{n \times d})$ on top of the attention score obtained from the soft prediction map:

$$\mathbf{A} = \mathrm{MSA}(\mathrm{BN}(\mathbf{z}))), \tag{7}$$

where $A_2$ shows the horizontally and vertically shifted version of the $A_1$ and $MSA$ stands for the multi-head self-attention. We then calculate the correlation matrix:

$$\rho_{A_1, A_2} = \frac{\mathrm{cov}\left(A_1, A_2\right)}{\sigma_{A_1} \sigma_{A_2}} = \frac{E\left[\left(A_1 - \mu_{A_1}\right)\left(A_2 - \mu_{A_2}\right)\right]}{\sigma_{A_1} \sigma_{A_2}}, \tag{8}$$

and seek to maximize the pair-wise correlation. To do so, we learn to minimize the $\mathcal{L}_1$-norm between the $\rho$ and Unit Matrix $I$. $\mathcal{L}_1$ is demonstrated to be a robust measure and causes fewer artifacts by being less disruptive to edges and linked regions in the image content. Therefore, we employ the $\mathcal{L}_1$-norm to calculate the differences in the score map to determine the spatial interactions between pixels and regions in the image:

$$\mathcal{L}_{intra}\left(\rho, I\right) = \sum_{i}^{n} \sum_{j}^{n} \|\rho_{i,j} - I_{i,j}\|_1 \tag{9}$$

Here, $\rho_{i,j}$ shows the pair-wise correlation value between the $i_{th}$ and the $j_{th}$ tokens in the score map $S$.

## 2.5. Joint objective

The final loss function utilized in our training process consists of three loss terms as follows:

$$\mathcal{L}_{\text{joint}} = \lambda_1 \mathcal{L}_{\text{ce}} + \lambda_2 \mathcal{L}_{\text{inter}} + \lambda_3 \mathcal{L}_{\text{intra}} \tag{10}$$

where the first term indicates the cross-entropy loss between the network's predicted scores and the maximum index to ensure prediction confidence and provide a mechanism to learn the distribution of each cluster. The second term shows the inter-scale loss which we included as a supervisory component to ensure feature representation stability across the scales and provide regions specific similarity. The final term is included to impose intra-scale consistency in each image region. This spatial consistency aims to reduce the local variation and provide a way to smoothly merge neighboring clusters.

## 3. Experiments

In our experiments, we conducted several ablation studies on two publicly available datasets as follows:

**Skin lesion segmentation**: As a first task, we evaluate our method on dermoscopic images to segment skin lesion regions. To this end, we use the PH2 dataset (Mendonça et al., 2013) that contains 200 RGB images of melanocytic lesions. The dataset covers a large variety of lesion types and demonstrates a real-world challenging problem. In our experiment, we use all 200 samples to evaluate the performance.

**Lung segmentation**: For the second task, we consider lung segmentation in CT images. We use the lung analysis dataset, which is publicly available by the Kaggle (see (Azad et al., 2019)) and offers 2D and 3D CT images. In our evaluation, we follow the same strategy presented in (Azad et al., 2019) to prepare the dataset. For the evaluation process, we extract the 2D slices from the 3D images and then select the first 700 samples to evaluate our method.

### 3.1. Experimental Set-up

**Network Details**: Our dual-branch structure consists of two efficient transformer blocks in each path to capture both fine-grained and global context representation. We use the batch-norm layer after each block to accelerate the model convergence. In our experiment, we set the small patch (e.g., equal to 1) and use a larger patch size (e.g., 2) in the second branch to model multi-scale representations. The intuition behind our selected patch sizes is to preserve spatial information at the pixel level while guiding the network through the large branch to model global dependency. This strategy ensures that the network captures the local representation and, at the same time, benefits from the regional information derived from the larger patch. As a result, we set patch sizes accordingly. For each image, we learn the trainable parameters by iteratively (maximum 50 iterations) minimizing the overall loss function using the SGD optimization with a learning rate of 0.1 and a momentum of 0.9. All experiments were performed on a single RTX 3090 GPU with the PyTorch library.

**Evaluation Protocol**: To set up a fair evaluation and provide more comparative insight into the performance of our suggested network, we use both unsupervised and self-supervised

clustering-based techniques to compare the effectiveness of our model. In this respect, using the Dice (DSC) score, XOR metric, and the Hammoud distance (HM), we compare our method with the unsupervised $k$-means clustering method and recent self-supervised methods, DeepCluster (Caron et al., 2018), IIC (Ji et al., 2019), and spatial guided self-supervised strategy (SGSCN) (Ahn et al., 2021). Following (Ahn et al., 2021), we only consider the cluster which has the higher overlap with the GT map as a target class prediction to evaluate our method. In our evaluation, the DSC score represents the agreement between the predicted target region and the GT map, with higher values indicating better results. On the contrary, HM and XOR metrics represent disagreement with the target; thus, lower values represent better performance.

### 3.2. Results

In this section, the performances of the proposed method against the SOTA approaches are presented. To begin with, we present comparative results on the skin lesion segmentation task in Table 1. Quantitative results show that our methods outperform the SOTA approaches in all metrics, which indicates the effectiveness of our strategy for self-supervised content clustering. Specifically, compared to SGSCN, our strategy models the spatial consistency at the token level, which further imposes a spatial dependency for the segmentation task. In addition, SGSCN minimizes the intra-cluster variation by imposing the context-based consistency on top of the score map, while our method applies the context agreement module to minimize the intra-cluster variation in the deeper level of the network, which effectively recalibrates the feature representation. From a qualitative perspective, we provide the comparative segmentation results on the skin lesion dataset in Figure 3. Unlike the DeepCluster and the k-means methods, it can be observed that our strategy produces a smooth segmentation map for the skin lesion area with a slightly better prediction of the lesion boundary. On top of that, SGSCN results in an under-segmentation map caused by considering edges around the lesion as a new class, while our method manages to achieve a slightly better segmentation map.

Moreover, the quantitative results of the suggested network on the lung region segmentation task are presented in Table 1. The noisy nature of CT images and the spiky ground truth labels usually limit the performance of even supervised methods for semantic segmentation on CT images. However, our self-supervised method produces acceptable results, which outperform the SOTA approaches in all metrics. Notably, the $k$-means algorithm performs well as the local areas in the lung dataset are quite similar to each other and have relatively simple shapes and fewer variations compared to skin lesions. Furthermore, we provide visual segmentation results in Figure 3 to illustrate the effectiveness of our approach. It can be seen that our method provides a softer contour of segmentation maps than other methods, which in our opinion, reflects that global long-range contextual information helped the network to perceive the actual boundary of the target and separate it from the background. Additionally, we analyze the effect of the suggested modules in Appendix A and the strength and limitations of our approach in Appendix B.

Table 1: Comparative performance of the proposed method against the SOTA approaches on the PH$^2$ and Lung datasets. Notably, $k$ is set to 3 in the PH$^2$ dataset and 2 in the Lung dataset.

| Methods | PH$^2$ | | | Lung Segmentation | | |
|---|---|---|---|---|---|---|
| | DSC ↑ | HM ↓ | XOR ↓ | DSC ↑ | HM ↓ | XOR ↓ |
| $k$-means | 71.3 | 130.8 | 41.3 | 92.7 | 10.6 | 12.6 |
| DeepCluster (Caron et al., 2018) | 79.6 | 35.8 | 31.3 | 87.5 | 16.1 | 18.8 |
| IIC (Ji et al., 2019) | 81.2 | 35.3 | 29.8 | - | - | - |
| SGSCN(Ahn et al., 2021) | 83.4 | 32.3 | 28.2 | 89.1 | 16.1 | 34.3 |
| **Our Method** | **86.0** | **23.1** | **25.9** | **94.6** | **8.1** | **14.8** |

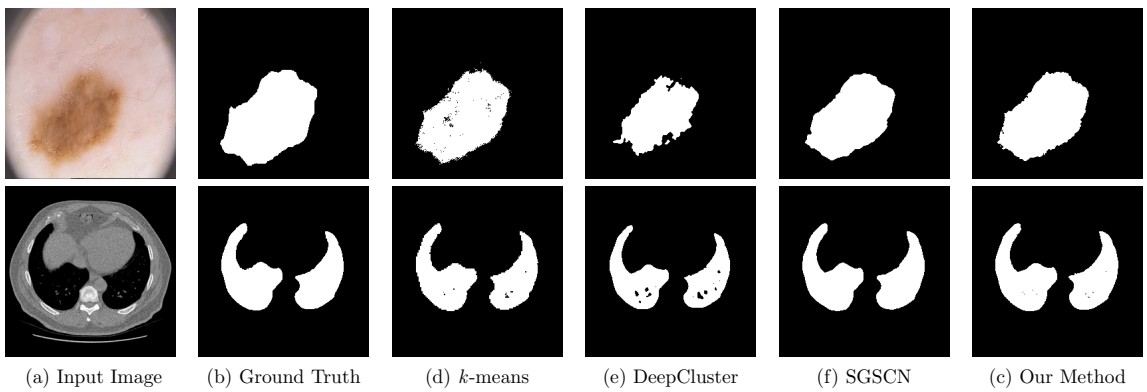

| (a) Input Image | (b) Ground Truth | (d) $k$-means | (e) DeepCluster | (f) SGSCN | (c) Our Method |
|---|---|---|---|---|---|

Figure 3: Visual comparison of different methods on the PH$^2$ skin lesion segmentation and Lung datasets. Our method generates segmentation maps with smoother boundaries and fewer false negative predictions compared to the DeepCluster approach. Additionally, our method achieves slightly better performance in delineating object boundaries compared to the SGSCN method.

## 4. Conclusion

In this paper, we proposed a self-supervised strategy to perform the medical image segmentation task without requiring any annotation mask. Our strategy builds upon an efficient self-attention mechanism in a dual-branch strategy to provide intra-scale and inter-scale consistency for clustering each pixel into a shared characteristic. In an iterative fashion, our algorithm produces semantically related segmentation maps which outperform the related SOTA approaches.

## 5. Acknowledgment

This work was funded by by the German Research Foundation (Deutsche Forschungsgemeinschaft, DFG) – project number 455548460.

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

## Appendix A. Ablation Study on the Effect of Suggested Modules

In our proposed method, we included the inter-scale and intra-scale feature consistency modules to further recalibrate the feature representation for a better clustering space. In this section, we first investigate the hyperparameter selection for the loss function weights and then we will elaborate on the effect of each module from both quantitative and qualitative perspectives to provide more insight into the contribution of these modules.

The hyperparameters of our proposed method were designed and tuned based on the empirical evaluation of the model on the small set of skin lesion segmentation images (10 samples) from ISIC 2017 (Codella et al., 2018). We used a grid search approach within a small range from (0 - 2)to explore the hyperparameter space and find the optimal values of $\lambda_1 = 1, \lambda_2 = 2, \lambda_3 = 1.2$. We use the obtained hyper-parameters for both datasets. To assess

the generalizability of the network, an additional experiment was conducted on the camera-ready version. The aim was to identify optimal hyperparameters for the lung segmentation dataset by utilizing ten samples from the aforementioned dataset. This resulted in the values of $\lambda_1 = 1.2$, $\lambda_2 = 1.8$, and $\lambda_3 = 1.2$. Subsequently, the updated hyperparameters were used to evaluate the performance of the model, and a slightly improved outcome was observed in comparison to the original configuration, with a Dice similarity coefficient (DSC) of 94.62. This approach highlights the importance of optimizing hyperparameters to achieve superior performance on a new dataset.

Moreover, in the inter-scale consistency mechanism, we included the second branch to capture long-range contextual information and provide a supervisory signal for inter-scale feature consistency. To scrutinize the exact contribution of the inter-scale module, we have conducted an experiment without including the inter-scale loss function. The results are presented in Table 2. It can be seen that, removing the inter-scale loss function resulted in a 1.9% DSC score loss compared to our main strategy. From a qualitative standpoint (Figure 4), we can also observe that by removing this loss function models tends to have difficulties in accurate boundary separation. In addition, it weakens the network multi-scale feature agreement condition and results in incorrectly merging the small cluster with the neighborhood clusters.

Table 2: Contribution of each loss function on the model performance. All experiments were performed on the PH$^2$ dataset.

| $\mathcal{L}_{ce}$ | $\mathcal{L}_{intra}$ | $\mathcal{L}_{inter}$ | DSC $\uparrow$ | HM $\downarrow$ | XOR $\downarrow$ |
|---|---|---|---|---|---|
| $\checkmark$ | $\times$ | $\times$ | 83.6 | 25.8 | 30.2 |
| $\checkmark$ | $\checkmark$ | $\times$ | 84.1 | 25.4 | 29.4 |
| $\checkmark$ | $\times$ | $\checkmark$ | 84.3 | 25.3 | 28.4 |
| $\checkmark$ | $\checkmark$ | $\checkmark$ | 86.0 | 23.1 | 25.9 |

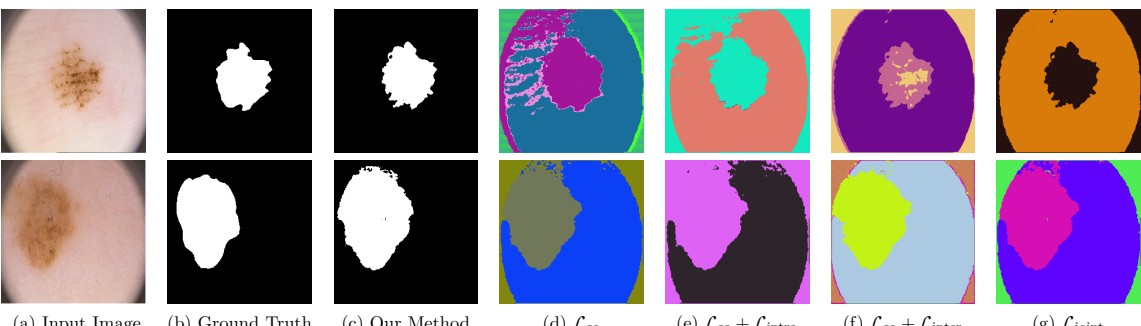

(a) Input Image   (b) Ground Truth   (c) Our Method   (d) $\mathcal{L}_{ce}$   (e) $\mathcal{L}_{ce} + \mathcal{L}_{intra}$   (f) $\mathcal{L}_{ce} + \mathcal{L}_{inter}$   (g) $\mathcal{L}_{joint}$

Figure 4: Segmentation results of the proposed method on the skin lesion segmentation task using the PH$^2$ dataset. (a) original input image, (b): grand truth map, (c-e) shows the obtained results by applying different loss combinations.

We investigated the effect of intra-scale consistency loss on the clustering process. Quantitative results are presented in Table 2. We can observe that our model without the intra-

scale loss function performs weakly in almost all metrics. This fact explains the importance of spatial consistency for the segmentation purpose. It is important to note that in our intra-scale consistency, we use shift size 2 (in x,y directions) to better model the local consistency. In fact, choosing a large shift step may result in a larger receptive field, which could lead to a loss of local consistency in the feature maps. This can result in wrong predictions and poor segmentation performance (as we observed in our experiments). Therefore, we chose a fixed shift step of 2 for both datasets to balance between gathering sufficient contextual information and preserving local consistency in the feature maps. Moreover, the visualization evidence (Figure 4) reveals that when the model does not impose the inter-scale (spatial) consistency the model tends to predict non-consistency clusters and lacks to perform cluster merging. Additionally, the importance of spatial consistency is more apparent when the algorithm deals with objects' surfaces. It is also worthwhile to mention that in our proposed method we modeled the intra-scale consistency loss using the correlation matrix $\rho$ between the contextual features calculated in the self-attention mechanism. To maximize the similarity we strived to minimize the $\mathcal{L}_1$-norm between the $\rho$ and Unit Matrix $I$. To visually investigate this process inside the deep model we visualized the correlation matrix during the training process in Figure 5. It is observable that the model learns to maximize the correlation in a $I$ form, which indicates the convergence of our spatial consistency.

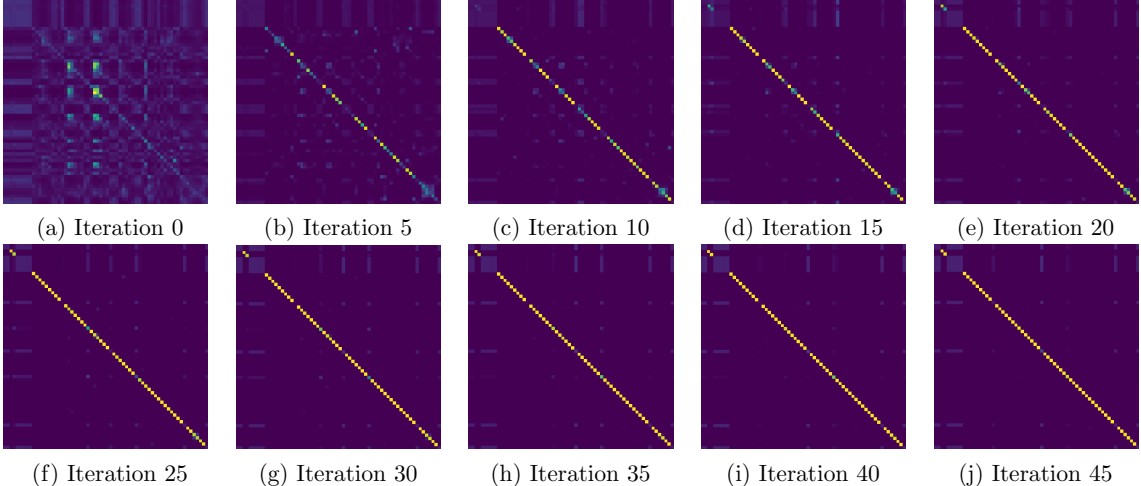

| (a) Iteration 0 | (b) Iteration 5 | (c) Iteration 10 | (d) Iteration 15 | (e) Iteration 20 |

| (f) Iteration 25 | (g) Iteration 30 | (h) Iteration 35 | (i) Iteration 40 | (j) Iteration 45 |

Figure 5: Convergence of the intra-scale correlation matrix through the training process.

It is worth mentioning that in our experiment, we set the $\lambda_1$, $\lambda_2$, and $\lambda_3$, in Equation 10, to 1, 5, and 1.2 for both datasets, respectively.

## Appendix B. Strength and Limitations

As we illustrated throughout the experimental results, our method surpasses the SOTA approaches on both datasets. Here we provide more visualization samples to investigate where our strategy can be more effective for self-supervised segmentation and in which cases our algorithm faces challenges. In this respect, we provided sample clustering results in Figure 6. Our method apparently produces a good clustering decision when there is

a sharp edge between the object boundaries. This might explain why our method usually produces slightly over-segmentation results. In Figure 7, we have provided some cases where our suggested method fails to predict the lesion regions. Judging from the qualitative results, it seems that our method renders a poor performance when the object of interest has a high overlap with background regions. More specifically, skin lesions may appear in a deformed shape which is quite challenging for the model to predict the exact lesion location. Besides that, the ground truth mask usually contains noisy annotation, which does not reflect the real segmentation map.

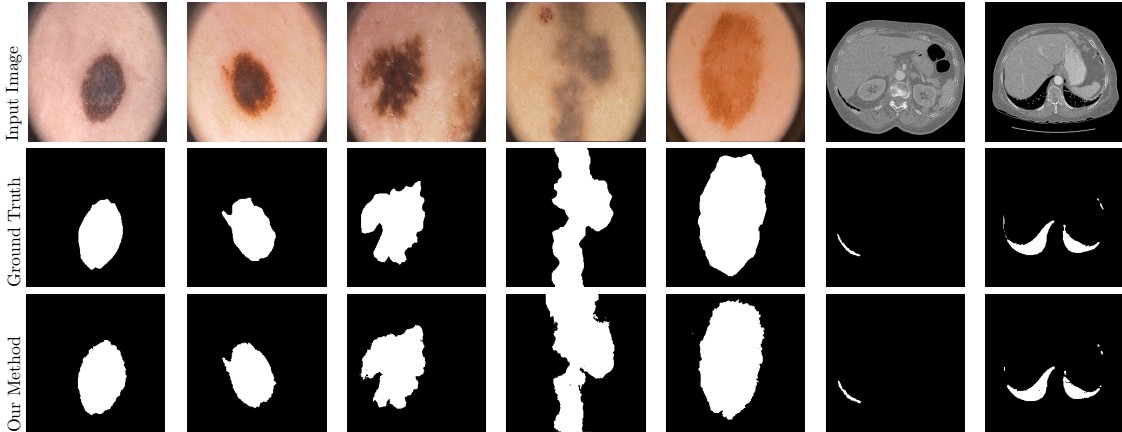

Figure 6: Some sample of the segmentation results in both PH$^2$ and Lung datasets.

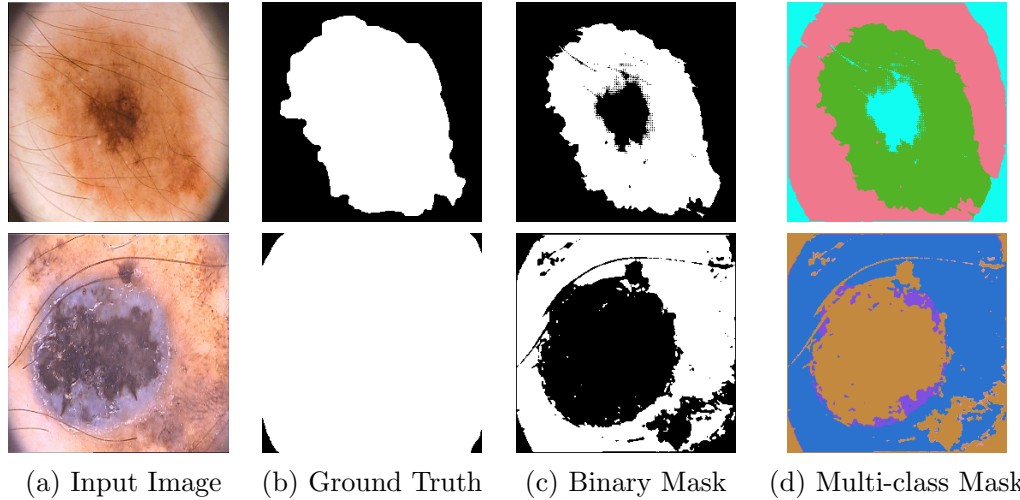

(a) Input Image     (b) Ground Truth     (c) Binary Mask     (d) Multi-class Mask

Figure 7: Sample limitation of the proposed method for skin lesion segmentation task.

