# OpenReview forum: "MS-Former: Multi-Scale Self-Guided Transformer for Medical Image Segmentation"
_MIDL.io/2023/Conference — MIDL 2023 Oral_

### Official Review · Reviewer_NK6K · 2023-02-01

**Confidence:** 5
**Preliminary Rating:** 4

**Summary:**

The paper proposed MS-Former, a self-supervised strategy to perform the medical image segmentation task without any annotations. The authors proposed a self-attention mechanism in a dual-branch strategy to provide intra-scale and inter-scale consistency for clustering each pixel into a shared characteristic. The proposed algorithm shows good performance compared to the SoTA methods on two benchmark datasets.

**Strengths:**

+ The proposed approach is novel and also achieves good segmentation results.
+ The paper is well-arranged and easy to read.
+ Experiment results also certify its effectiveness.
+ Detailed ablations are presented to show the effectiveness of MS-Former.


**Weaknesses:**

+ The setting of hyper-parameters is needed to explain in detail. How to design them? How about the detailed tuning?
+ Results of k-means have large differences on two medical datasets, why?
+ Although the overall architecture is novel, its individual components are largely inspired by previous works: (1) transformer module [1]; (2) self-supervised learning [2,3], (3) semi-supervised learning - consistency loss [4,5]. If these studies have any relevance to the topic at hand, it would be great if the authors would highlight them.

Reference:

[1] Class-Aware Adversarial Transformers for Medical Image Segmentation

[2] Bootstrapping semi-supervised medical image segmentation with anatomical-aware contrastive distillation

[3] Mine your own anatomy: Revisiting medical image segmentation with extremely limited labels

[4] Momentum Contrastive Voxel-wise Representation Distillation for Semi-supervised Volumetric Medical Image Segmentation

[5] SimCVD: Simple contrastive voxel-wise representation distillation for semi-supervised medical image segmentation

**Deanonymize Review:**

no

**Paper Type:**

methodological development

**Questions To Address In The Rebuttal:**

+ The setting of hyper-parameters is needed to explain in detail. How to design them? How about the detailed tuning?
+ Results of k-means have large differences on two medical datasets, why?
+ Although the overall architecture is novel, its individual components are largely inspired by previous works: (1) transformer module [1]; (2) self-supervised learning [2,3], (3) semi-supervised learning - consistency loss [4,5]. If these studies have any relevance to the topic at hand, it would be great if the authors would highlight them.

Reference:

[1] Class-Aware Adversarial Transformers for Medical Image Segmentation

[2] Bootstrapping semi-supervised medical image segmentation with anatomical-aware contrastive distillation

[3] Mine your own anatomy: Revisiting medical image segmentation with extremely limited labels

[4] Momentum Contrastive Voxel-wise Representation Distillation for Semi-supervised Volumetric Medical Image Segmentation

[5] SimCVD: Simple contrastive voxel-wise representation distillation for semi-supervised medical image segmentation

---

### Official Review · Reviewer_DX35 · 2023-02-03

**Confidence:** 4
**Preliminary Rating:** 4
**Recommendation:** Poster

**Summary:**

A novel multi-scale segmentation solution is proposed that uses a self-guided transformer. The authors introduce both an intra- and an inter-scale consistency term to improve on the clustering of similar pixels. The framework is tested on two very different types of cohorts: a skin lesion and the lung CT datasets. The performance of the new tool is demonstrated both qualitative and quantitatively, by comparing to both unsupervised and self-supervised solutions.

**Strengths:**

The submission is clearly written. It well describes the literature and the proposed solution.

The proposed multi-scale transformer framework is innovative and the experimental section provides encouraging outcomes on two very different datasets.

Superior performance was demonstrated compared to some state-of-the-art solutions.

The paper addresses the interesting problem of learning annotations when no previously segmented examples exist.






**Weaknesses:**

I suspect that this is a 2D solution as of now, but it is not clear. Can you confirm? The luncg CT cohort does have 3D images, but it is not clear that those were used.

\ro in Eq 1 is not defined.

The PH2 data set was not introduced. Could more detail be included in the submission about it?


**Deanonymize Review:**

no

**Detailed Comments:**

Could the authors reorder the results from the various tools in the plots / figures? Can you have sgscn next to the proposed tools for easier comparison, for example?

**Paper Type:**

both

**Questions To Address In The Rebuttal:**

How easy would it be to expand / run this framework in 3D, given there are already memory limitation that were mentioned.

Can authors expand the discussion of the results? How would they explain, for example, that for the lung k-means is almost as good as the proposed method?

---

### Official Review · Reviewer_bLYD · 2023-02-10

**Confidence:** 3
**Preliminary Rating:** 4
**Recommendation:** Poster

**Summary:**

The author proposed a multi-scale self-supervised based segmentation method and surpassed SOTA methods in two public datasets according to the experiments. No segmentation masks are required for this segmentation method. Specifically, an efficient Transformer was selected as the backbone. In addition to the cross-entropy loss for clustering-based segmentation, another two multi-scale consistency constraints were designed to further improve the segmentation performance.

**Strengths:**

1. The paper is well-written and easy to follow.
2. Propose an inter-scale and an intra-scale consistency loss. The ablation study with visualization shows the effectiveness of proposed consistency losses.
3. Surpassed SOTA in two public datasets.
4. Ablation studies and failure cases had been provided.
5. Code has been released.

**Weaknesses:**

1. It seems that the proposed unsupervised method may fail in some complex segmentation tasks since the original softmax prediction may be totally wrong and hard to correct/improve by the cross entropy loss (between softmax and argmax results). It is expected to clarify the restriction of datasets.


**Deanonymize Review:**

no

**Detailed Comments:**

1. It seems that the SGSCN and the proposed method achieved similar performance in figure 3. Since the boundary of the mask is discussed, it is probably better to visualize the comparison by contours instead of masks.
2. It is better to explain "MSA" before using the abbreviations directly.
3. It would be interesting to see the comparison of the proposed self-supervised version to the supervised version.

**Paper Type:**

both

**Questions To Address In The Rebuttal:**

1. Why the number of training data and data splits are not reported? Why there is no performance for Lung segmentation of the IIC method?
2. In Intra-scale Consistency, how large the shift step is? It is better to explain more about the A2.

---

### Meta-Review · Area_Chair_xNz6 · 2023-02-20

**Recommendation:** Accept (Poster)
**Confidence:** 5

**Metareview:**

The paper proposes a novel multi-scale segmentation solution that uses a self-guided transformer and introduces both an intra- and an inter-scale consistency term to improve clustering of similar pixels. The paper is well-written and easy to follow and the proposed approach achieves good segmentation results on two different datasets, surpassing some state-of-the-art solutions. The paper also addresses the interesting problem of learning annotations when no previously segmented examples exist. Ablation studies and failure cases are provided and the code has been released.

Strengths: The proposed approach is novel and achieves good segmentation results. The paper is well-arranged and easy to read. Detailed ablations are presented to show the effectiveness of the proposed approach. The paper addresses an interesting problem and provides a solution that can learn annotations when no previously segmented examples exist. Ablation studies and failure cases are provided and the code has been released.

Weaknesses: The setting of hyper-parameters is not explained in detail. The results of k-means have large differences on two medical datasets, and it is not clear why. Although the overall architecture is novel, its individual components are largely inspired by previous works. It is expected to clarify the restriction of datasets, and whether the proposed unsupervised method may fail in some complex segmentation tasks.

After the rebuttal, I think most of the concerns from the reviewers have been addressed.